# Utilization of Steel Slag in Blind Inlets for Dissolved Phosphorus Removal

**Javier M. Gonzalez \*, Chad J. Penn and Stan J. Livingston**

National Soil Erosion Research Laboratory, USDA-ARS, West Lafayette, IN 47907, USA;
chad.penn@usda.gov (C.J.P.); stan.livingston@usda.gov (S.J.L.)

\* Correspondence: javier.gonzalez@usda.gov

**Abstract:** Blind inlets are implemented to promote obstruction-free surface drainage of field depressions as an alternative to tile risers for the removal of sediment and particulate phosphorus (P) through an aggregate bed. However, conventional limestone used in blind inlets does not remove dissolved P, which is a stronger eutrophication agent than particulate P. Steel slag has been suggested as an alternative to limestone in blind inlets for removing dissolved P. The objectives of this study were to construct a blind inlet with steel slag and evaluate its ability to remove dissolved P, nitrogen (N), and herbicides. A blind inlet was constructed with steel slag in late 2015; data from only 2018 are reported due to inflow sampling issues. The blind inlet removed at least 45% of the dissolved P load and was still effective after three years. The dissolved P removal efficiency was greater with higher inflow P concentrations. More than 70% of glyphosate and its metabolite, and dicamba were removed. Total N was removed in the form of organic N and ammonium, although N cycling processes within the blind inlet appeared to produce nitrate. Higher dissolved atrazine and organic carbon loads were measured in outflow than inflow, likely due to the deposition of sediment-bound particulate forms not measured in inflow, which then solubilized with time. At a cost similar to local aggregate, steel slag in blind inlets represents a simple update for improving dissolved P removal.

**Keywords:** blind inlet; dissolved P; steel slag; atrazine; glyphosate; nitrogen; dissolved C; water quality

## 1. Introduction

Tile risers are vertical perforated pipes used for draining farmed closed depressions in the U.S. Midwest. Although this practice is effective for its intended purpose, sediment and pollutants may be transported more directly through tile risers and ultimately into waterways, causing environmental concerns. An alternative to the tile riser is the blind inlet, a conservation practice proven for draining farmed closed depressions with added benefits, e.g., filtering surface runoff sediments and pollutants, including particulate P (phosphorus) and pesticides [1–3]. A blind inlet is constructed by excavating soil from a depression, placement of a layer of a perforated drainage pipe that is connected to a tile drain, and back-filling with highly permeable aggregate to the surface [4]. One advantage of the blind inlet over tile risers is that they do not interfere with field management activities, such as planting and harvest. However, some soluble compounds, e.g., dissolved P, are not retained by traditional blind inlets [4,5]. While retention of particulate P via sediment capture is beneficial, the loss of dissolved P is considered more problematic in some watersheds, such as the Western Lake Erie Basin. Dissolved P losses to surface waters are not only more difficult to control but due to the immediate bioavailability of dissolved P to aquatic life, it is a stronger eutrophication agent than particulate P.

The implementation of blind inlets as a conservation practice is relatively new. Typically, readily available local aggregates (e.g., sand and gravel) are used to construct blind inlets. In the U.S. Midwest,

limestone has been used as the primary blind inlet bed material, which is effective for sediment retention [1,3]. In 12 years, a blind inlet with limestone removed about 40% sediment and particulate P [4]. However, limestone is a poor sorbent for removing dissolved constituents, such as P or certain herbicides [6]. With flow-through experiments, it was observed that the ability of limestone sand and gravel to remove dissolved P was limited [4]. Hence, alternative blind inlet bed materials that retain dissolved P and other contaminants are needed to improve the capabilities of the blind inlet as a conservation practice for draining closed depressions. Over the last two decades, several P sorption materials (PSMs) for removing dissolved P from flowing water have been studied and implemented in P removal structures [7,8], which are landscape-scale filters with PSMs designed to remove dissolved P from non-point drainage water before reaching a surface water body. While diverse in appearance, P removal structures are designed to contain sufficient PSM mass to remove a portion of the annual dissolved P load at a site, to handle typical flow rates for a site, and allow for PSM replacement after it is no longer effective [9]. The replacement of traditional limestone sand and gravel with a PSM would allow the blind inlet to serve as a P removal structure [4]. Many PSMs are industrial by-products, such as steel slag, flue gas gypsum, mine drainage residuals, steel turnings, and other materials rich in Al/Fe oxides/hydroxides or highly soluble Ca minerals [7,10]. Due to the low solubility of P on PSMs that have been previously used for P removal, they are mostly unable to be utilized a P fertilizer. However, depending on the PSM characteristics, they may still provide some benefit through land application, such as S and Ca additions. For spent steel slag, the most common disposal/re-use method is as a construction aggregate, for example, secondary roads. This is logical since the most common use of non-spent slag is as a construction aggregate and custom sieved to meet a desired particle size distribution, where much of it is sand and gravel size. In addition, steel slag is abundant and inexpensive in the US Midwest, where blind inlets are mostly installed. Due to its availability, ability to be used as a construction aggregate, high permeability, and P sorption capacity, steel slag is a logical alternative to limestone gravel for efficiently improving the ability of blind inlets to remove dissolved P.

Little is known about how steel slag material might impact the removal of other water pollutants, including herbicides. In the only study known, using sorption batch isotherms, an electric-arc-furnace (EAF) slag material removed only 0.6% of the atrazine in solution. In contrast, limestone and oak (*Quercus* spp.)-derived biochar removed 6.7% and 99.0%, respectively [6]. The low sorption of atrazine by EAF slag was attributed to its high pH since atrazine sorption is pH dependent [6,11]. No information is available on the potential for slag to remove other herbicides commonly used in U.S. Midwest agriculture systems. The objectives of this study were to construct and evaluate a blind inlet using EAF slag as an alternative to traditional limestone to (1) improve dissolved P removal, and (2) determine the potential impact of EAF slag on herbicide removal.

## 2. Materials and Methods

### 2.1. Site Description and Management

The study site is in northeast Indiana, USA (41°26′26.65″ N, 84°56′56.18″ W). The topography is very gently rolling upland till plain, and the primary soils are an association of Glenwood-Blount series. The site has been mostly under no-till with corn (*Zea mays*)/soybean (*Glycine max*) rotation and winter cover crops (a mixture of *Secale cereale* and *Hordeum vulgare*) implemented after harvest of the corn or soybean. In the fall of 2015, new tiles were installed, followed by the entire field being disked. The site was planted with corn on 26 May 2018 and harvested on 5 December 2018. Other inputs (including herbicides and fertilizers) applied to the site are listed in Table 1.

**Table 1.** Field management practices implemented at the field site during the blind inlet monitoring period, regarding the application of chemicals, nutrients, and planting and harvest activities.

| Date | Management | Rate (kg ha$^{-1}$) |
|---|---|---|
| 5 June 2017 | Soybean Planting | |
| 5 October 2017 | Ammonium-N | 10 |
| 5 October 2017 | Sulfate-S | 24 |
| 5 October 2017 | Potassium | 186 |
| 5 October 2017 | Gypsum Sulfate-S | 53 |
| 5 October 2017 | Gypsum Ca | 66 |
| 20 October 2017 | Soybean harvest | |
| 16 March 2018 | Potassium | 186 |
| 16 March 2018 | Ammonium-N | 28 |
| 16 March 2018 | Sulfate-S | 63 |
| 16 March 2018 | Chicken litter | 4477 |
| 1 May 2018 | Glyphosate | 1.68 |
| 1 May 2018 | Dicamba | 0.38 |
| 26 May 2018 | Corn Planting | |
| 8 June 2018 | Glyphosate | 1.05 |
| 8 June 2018 | Atrazine | 1.58 |
| 26 June 2018 | Nitrogen (28-0-0) | 140 |
| 5 December 2018 | Corn Harvest | |

*2.2. Slag Source, Preparation, Characterization*

The slag material was from an electric-arc furnace (EAF) steel mill in Butler, Indiana, USA. Two different EAF slag sizes were obtained, the fine (1 to 5 mm) and coarse (10 to <50 mm) fractions. The coarse size fraction was considered inert and used for the purpose of physical drainage, not chemical P removal. For the remainder of this paper, unless specified otherwise, all references to slag in the blind inlet and for laboratory analysis is assumed to be of the "fine" 1–5 mm fraction. Slag was sieved at the steel plant by the slag-handling company (Edwin Levy Co., Dearborn, MI, USA). After delivery to the site, a subsample of EAF slag was taken for physical and chemical characterization. Briefly, particle size distribution was determined via sieve analysis and bulk density was measured by weighing a known volume of slag. Total elemental analysis of the EAF slag was achieved using heated acid digestion and followed by analysis of the digestate by inductively coupled plasma atomic emission spectroscopy (ICP-AES) for Ca, Mg, Fe, Al, Mn, K, Cr, Ni, Pb, Zn, Cu, and S [12]. pH and electrical conductivity (EC) were measured in de-ionized (DI) water with a pH and EC meter after equilibrating slag for 30 min in a 1:5 solid/solution.

Slag was tested for P removal under lab conditions using a flow-through cell [7,9]. Briefly, two tests were conducted in duplicate, in which 0.2 and 2.5 mg P L$^{-1}$ was supplied via a Mariotte bottle to a flow-through cell containing 2 g of slag. The flow rate was adjusted to achieve a 10-min retention time (retention time = pore volume divided by flow rate) using a peristaltic pump located downstream/below the cell. Outflow samples were collected every 2 h and analyzed for dissolved P using the colorimetric method described in a later section. Inflow concentrations of 2.5 and 0.2 mg L$^{-1}$ were chosen to represent the magnitude of P concentration from runoff when poultry litter was recently applied, and also after soil equilibration, as observed in field data.

*2.3. Design and Construction of the Blind Inlet*

The blind inlet was constructed in the late fall of 2015 following the USDA-NRCS guidelines for the Conservation Practice Underground Outlet Code 620 [13]. The blind inlet measured 4.57 m × 4.57 m × 0.91 m (W × L × D). Geotextile fabric was placed on the bottom and sides of the excavation. Approximately 0.05-m of coarse EAF slag was placed on the geotextile, followed by a collection grid consisting of a perimeter (3.05 m × 3.05 m) of perforated plastic pipe (polyvinyl chloride; 10 cm diameter). One additional section of pipe was connected from one side of the collector to the

other, oriented towards the outlet and connected to a solid outlet pipe (Figure 1) before 0.56 m was back filled with the coarse EAF slag material. Geotextile fabric was placed on top of the course layer, and the remaining upper 0.30 m was back filled with the fine EAF slag material. The mass of fine slag in the upper layer was approximately 15.2 Mg. The blind inlet was located within 25 m of a ditch, where the solid outlet pipe discharged.

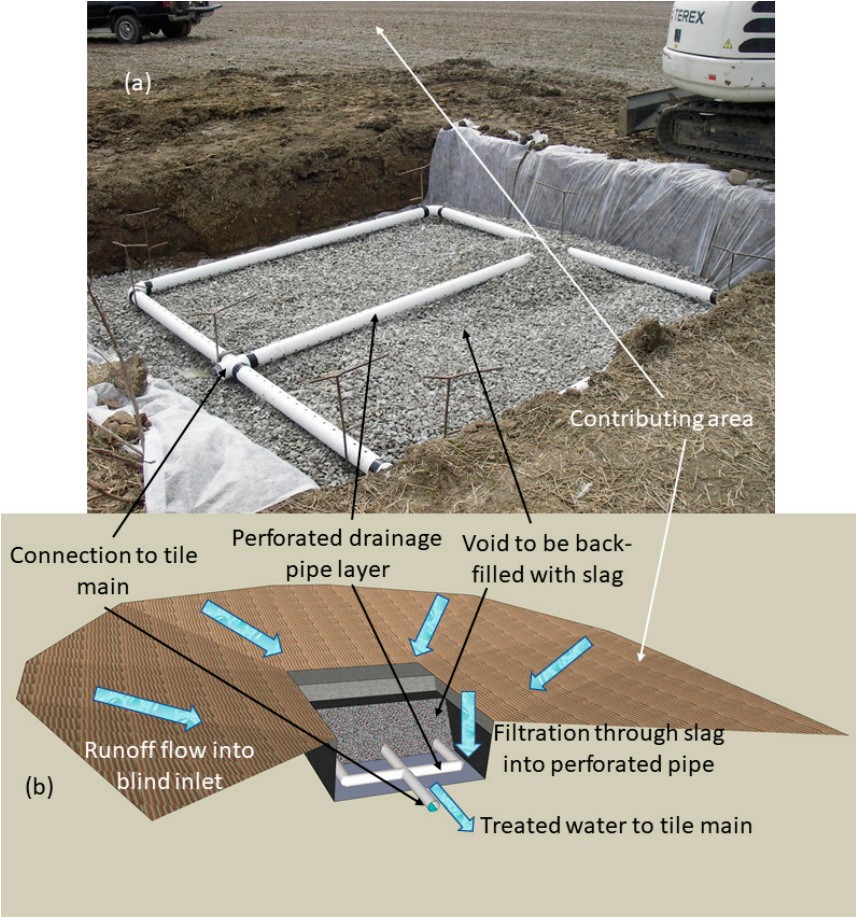

**Figure 1.** Photograph of the blind inlet during construction, specifically of the drainage pipe before being connected to the tile main and before final back-filling with steel slag (**a**), and diagram of the blind inlet constructed with steel slag for enhanced dissolved phosphorus (P) removal (**b**).

*2.4. Monitoring of the Blind Inlet*

The outlet pipe of the blind inlet served as a monitoring and sampling location at the point of discharge into the ditch. Flow rate was measured using an ISCO TIENet 350 Area Velocity sensor (Teledyne ISCO, Lincoln, NE, USA) for high flow and a Thel-mar Weir (Thel-Mar, LLC, Brevard, NC, USA) for low flow. Runoff events were triggered based on a threshold water level above the surface of the blind inlet, as measured by an ISCO TIENet 330 bubbler. Discrete 300-mL water samples were taken by an ISCO 6712 autosampler every 30 min, starting when the surface water was above the surface of the blind inlet. The following thresholds were required for the autosampler to begin collecting samples: Water level > 20 mm above the blind inlet surface for 5 min and the level of flow exiting the blind inlet > 85 mm for 10 min. Autosamplers were temperature controlled to maintain 4 °C. Due to the capacity of the autosampler and frequency of sample collection, the maximum number of samples per runoff event was 24. After each runoff event, samples were collected and processed (see the section below) in a satellite laboratory, followed by storage at −4 °C until being transported to a central lab until analysis.

## 2.5. Chemical Analysis of Water Samples

Water samples were vacuum-filtered through 0.45-µm nylon filters with a 20-mL aliquot acidified with 100 µL of 50% $H_2SO_4$ solution. These samples were used to determine dissolved P, $NH_4$-N, and $NO_3$-N colorimetrically using a Thermo Scientific Gallery autoanalyzer (Thermo Scientific, Waltham, MA, USA) following USEPA methods 365.2, 350.1 rev 2, and 353.1, respectively [14]; the limits of detection were 0.005, 0.01, and 0.01 mg $L^{-1}$, respectively.

Total Kjeldahl nitrogen (TKN) and total phosphorus (TP) were determined in unfiltered, acidified (60 mL sample + 100 µL of 50% $H_2SO_4$ solution) water samples following EPA methods 351.2 rev. 2 and 365.4, respectively [14]. Briefly, samples were digested with mercuric sulfate on a Lachat BD-46 block digestor (Hach, Loveland, CO, USA). Then, TKN and TP were determined in a Thermo Fisher Konelab colorimetric autoanalyzer (Thermo Scientific, Waltham, MA, USA). The limits of detection for both TKN and TP were 0.01 mg $L^{-1}$.

Dissolved C content was determined in filtered (0.45-µm nylon filters) and non-acidified samples using wet chemical oxidation and non-dispersive infrared detection of $CO_2$ with a Shimadzu TOC-Vws analyzer (Shimadzu Inc., Kyoto, Japan). Total dissolved C (TDC) in the sample was determined by oxidizing all C with sodium persulfate and phosphoric acid to produce $CO_2$. Inorganic C (IC) content was determined by adding phosphoric acid to the sample to produce $CO_2$. Potassium hydrogen phthalate and sodium bicarbonate were used as external standards (range from 5 to 80 mg $L^{-1}$) for TDC and IC, respectively. The dissolved organic C (DOC) content was calculated as the difference between TDC and IC.

Soluble $K^+$, $Mg^{2+}$, $Ca^{2+}$, and $SO_4^{2-}$-S were determined in unacidified syringe-filtered (0.45-µm nylon filter) water samples using two 2100 Dionex ion chromatography systems (Thermo Scientific, Waltham, MA, USA) equipped with eluent generators and electrical conductivity detectors. An ion chromatography system was dedicated to the analysis of cations and the other for anions, where these two systems shared an AS-AP ICS-5000 autosampler (50-µL sample injection for each). A 250 mm length × 4 mm diameter Dionex IonPac CS12A ion column was used to separate the cations with isocratic 20 mM methanesulfonic acid as eluent with a flow rate of 1 mL $min^{-1}$. After the separation of cations, the eluent current was suppressed with a CSRS 300 4-mm suppressor (Thermo Scientific, Waltham, MA, USA) set at 59 mA. Then, the electrical current of each cation was measured with a DS6 Dionex electrical conductivity cell set at 30 °C. Identification and quantification of cations were performed using a mixture of external standards. Calibration curves were performed with concentrations ranging from 0.06 to 40 mg $L^{-1}$ and using the 1/concentration weighting linear curve fitting. Retention times of $K^+$, $Mg^{2+}$, and $Ca^{2+}$ were 6.73, 10.37, and 13.51 min, respectively. Separation of anions was performed with a 250 mm length × 4 mm diameter Dionex IonPac AS18 column set at 40 °C and a KOH solution gradient as eluent (10 mM hold from the 1 min, ramp to 35 mM for 8 min, ramp to 45 mM for 1 min and hold for 5 min, then 10 mM for 3 min). The flow rate of the eluent was 1.2 mL $min^{-1}$. An ASRS-ULTRA II 4-mm suppressor (Thermo Scientific, Waltham, MA, USA), set at 134 mA, was used to suppress the electrical current of the eluent in the anion analysis. The electrical conductivity cell was set at 40 °C to measure the electrical current of the anions. Like the cations, external standards (0.05 to 40 mg $L^{-1}$) were used to perform calibration curves with 1/concentration weighting linear curve fitting and to quantify the amount of anions. Retention time of $SO_4^{2-}$-S was 8.79 min.

The herbicide atrazine and its metabolites deethylatrazine (DEA), deisopropylatrazine (DIP), and 2-hydroxyatrazine (OH-ATZ) were analyzed in unacidified syringe-filtered (0.45-µm nylon filter) water samples using a Waters Acquity ultra-performance liquid chromatography (UPLC) system (Milford, MA, USA) coupled with a Waters Acquity TQ tandem quadrupole mass (MS) detector. The separation of the compounds was performed with a 100 mm × 2.1 mm × 1.7 µm Waters Acquity UPLC BEH C18 column using a mobile phase gradient with 0.01% formic acid (A) and acetonitrile (B) and flow rate of 0.45 mL $min^{-1}$. The initial gradient was: 30% B, hold for 0.70 min, increase to 60% B in 4.8 min, increase to 75% B in 0.5 min, hold for 0.5 min, decrease to 30% B in 0.5 min, and hold for

0.5 min. All solvents were Optima grade (Thermo Fisher Scientific Inc., Waltham, MA, USA). The MS detector was set in positive ionization, and the Multiple Reaction Monitoring mode was used for the detection and confirmation of compounds. The MS voltages in the capillary, cone, extractor, and radio frequency lens were 0.61, 40, 3.0, and 0.1 kV, respectively. Source and desolvation temperatures were set at 150 and 400 °C, respectively. Desolvation and cone gas flows were set at 850 and 20 L h$^{-1}$, respectively. For each compound, the mass-to-charge ratio (m/z) of the parent material was used for quantification, and the most prominent fragment was used for confirmation of the compound: Atrazine: 216 > 156 > 174; DEA: 188 > 146 > 104; DIP: 174 > 104 > 132; OH-ATZ: 198 > 156 > 86. External standards (0.05 to 40 µg L$^{-1}$) were prepared in 25% methanol. The calibration curve was forced to the origin, fitted to the quadratic form, and 1/concentration as the weighted function. To minimize introduction of salts from the samples to the MS detector, samples were diluted as follows: 0.5 mL of sample + 0.45 mL of nanopure water + 0.05 mL of 100 µg L$^{-1}$ D5-atrazine (as internal standard) were added to a 1.5-mL vial and vortexed before analysis. The internal standard (m/z quantification and confirmation 221 > 101 > 69) was also added to the external standards. The retention times of atrazine, D5-atrazine, DEA, DIP, and OH-ATZ were 2.90, 2.90, 1.20, 0.93, and 0.77 min, respectively.

Dicamba and glyphosate (and its metabolite aminomethyl phosphonic acid, AMPA) were analyzed by ion chromatography as anions using the same method for $SO_4^{2-}$-S; the retention times were 15.33, 12.70, and 7.05 min, respectively. The ranges of the calibration curves were from 0.01 to 5.00 mg L$^{-1}$ for dicamba, from 0.025 to 5.00 mg L$^{-1}$ for glyphosate, and 0.10 to 5.00 mg L$^{-1}$ for AMPA.

## 2.6. Calculations and Data Analysis

Since water samples were collected at 30-min intervals during runoff events, the 30-min inflow and outflow discharge volumes were calculated by multiplying the measured water flow rate (L s$^{-1}$) by the elapsed time of 30 min. The 30-min mass of an analyte was then calculated by multiplying the 30-min discharge volume by the concentration of the analyte (mg L$^{-1}$ for nutrients and dissolved C or µg L$^{-1}$ for herbicides). Inflow and outflow volume and analyte mass for each runoff event were calculated by summing the 30-min values for each runoff event. Cumulative inflow and outflow values were calculated by summing the discharge volume and analyte mass of all runoff events. The flow-weighed mean concentration (FWMC) of each analyte per runoff event for the inflow and outflow was calculated by dividing the mass of the analyte by the runoff event discharge volume. The percent removal of dissolved P by the EAF slag material was calculated as follows:

$$\% \text{ Removal of Dissolved P} = \frac{(\text{Inflow mass P} - \text{ Outflow mass P})}{\text{Inflow mass P}} \times 100 \tag{1}$$

where Inflow and Outflow mass P is per runoff event or sum of all the event P (in kg).

## 3. Results and Discussion

### 3.1. Site Management, Runoff Production, and Slag Characterization

The collection of inlet pre-treated samples was either incomplete or absent for 2016–2017, resulting in incomplete data sets for that period. Repairs to the inlet collection sampling were made in late 2017; therefore, only runoff event data from 2018 are presented in this study. A total of seven runoff events were distributed from 3 April to 31 December and occurred with precipitation ranging from 12 to 46 mm per runoff event (Table 2). Runoff events # 1 and 2 (3 and 15 April) occurred 19 and 31 days, respectively, after the application of poultry litter (Table 1). Runoff event # 3 (3 May) occurred 2 days after application of glyphosate and dicamba. Runoff event # 4 (22 June) occurred 4 days after the application of urea and 15 days after the application of atrazine and the second application of glyphosate.

**Table 2.** Precipitation and temperature for the runoff events monitored at the blind inlet.

| Date | Runoff Event # | Cumulative Precipitation [1] | Precipitation Rate [1] | Max Temp [2] | Min Temp [2] | Average Temp [2] |
|---|---|---|---|---|---|---|
| | | mm | mm h$^{-1}$ | °C | | |
| 3 April | 1 | 18 | 2.48 | 15.2 | −0.1 | 2.6 |
| 15 April | 2 | 27 | 1.34 | 10.0 | −0.1 | 3.8 |
| 3 May | 3 | 38 | 0.84 | 29.2 | 15.4 | 20.4 |
| 20 June | 4 | 25 | 1.06 | 19.9 | 15.8 | 18.0 |
| 1 November | 5 | 30 | 1.45 | 8.8 | 2.7 | 5.1 |
| 26 November | 6 | 12 | 1.07 | 3.9 | −1.7 | 0.0 |
| 31 December | 7 | 46 | 1.67 | 10.6 | −2.4 | 2.7 |

[1] Cumulative precipitation and average precipitation rate per runoff event. [2] Maximum, minimum, and average air temperature per runoff event.

Steel slag chemical properties were typical for material produced from the EAF steelmaking process (Table 3). Slag had an elevated pH and high Ca concentration, both of which are required for the slag to be effective at dissolved P removal via precipitation of Ca phosphate [7,15–18]. Although slag contained some trace metals, there was no environmental concern about release to treated water since EAF slag has been shown to possess negligible metal solubility [9,19] and even may remove trace metals [20]. Flow-through analysis (Figure 2) revealed typical P removal for a slag sample mostly devoid of fines less than 1 mm in diameter [19,21,22]. The higher removal of dissolved P at 2.5 mg L$^{-1}$, relative to 0.2 mg L$^{-1}$, by the EAF slag (Figure 2) is typical for EAF slags and other PSMs dominated by Ca-P precipitation [7,23]. Higher P, together with high Ca concentrations and elevated pH environments, provide the chemical conditions for the Ca-P precipitation reactions to occur. Thus, runoff water with high dissolved P concentrations is ideal for the construction of P removal structures [8,9].

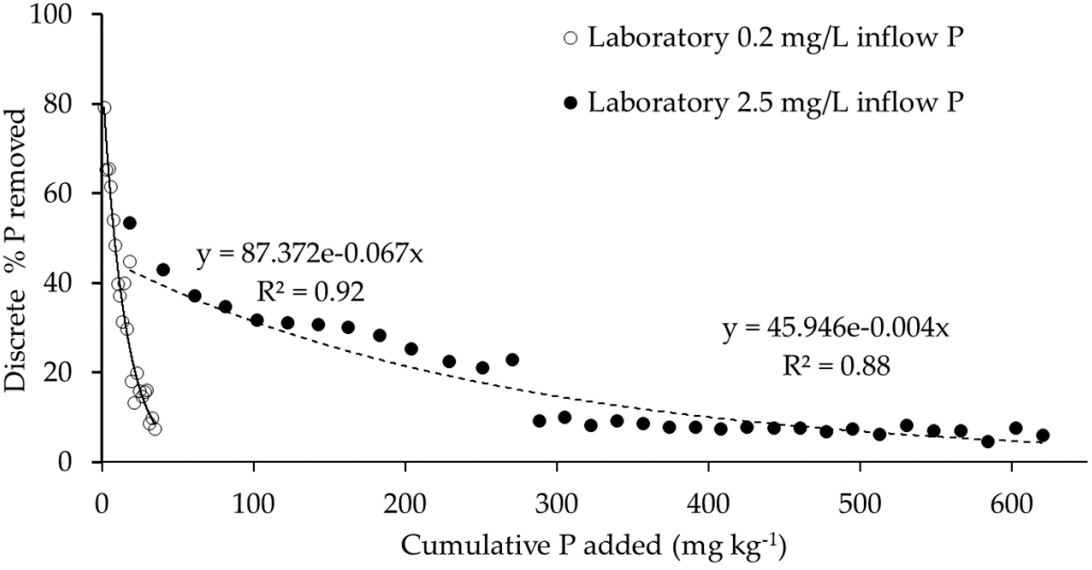

**Figure 2.** Discrete dissolved phosphorus (P) removed by the slag under laboratory flow-through conditions with a retention time of 10 min and inflow concentrations of 0.2 and 2.5 mg L$^{-1}$, expressed as a function of cumulative P added per kg slag. Both relationships were significant at *p* < 0.05.

**Table 3.** Physical and chemical properties of the slag used in the blind inlet and laboratory flow-through experiments. Concentrations of elements are total values.

| Parameter | Units | Value |
|---|---|---|
| Physical properties | | |
| >6.4–9.3 mm | % | 0.1 |
| 4.8–6.4 mm | % | 5.2 |
| 2.4–4.8 mm | % | 88.0 |
| 1.2–2.4 mm | % | 4.8 |
| 0.6–1.2 mm | % | 0 |
| 0.3–0.6 mm | % | 0.2 |
| 0.15–0.3 mm | % | 0.3 |
| 0.075–0.15 mm | % | 0.3 |
| <0.0075 mm | % | 1.1 |
| Bulk density | g cm$^{-3}$ | 1.70 |
| Chemical properties | | |
| pH | | 10.09 |
| Electrical conductivity | mS cm$^{-1}$ | 0.51 |
| Ca | g kg$^{-1}$ | 175 |
| Mg | g kg$^{-1}$ | 74.4 |
| Fe | g kg$^{-1}$ | 321 |
| Al | g kg$^{-1}$ | 18.1 |
| Mn | g kg$^{-1}$ | 42.3 |
| K | mg kg$^{-1}$ | 104 |
| Cr | mg kg$^{-1}$ | 4540 |
| Ni | mg kg$^{-1}$ | 6.39 |
| Pb | mg kg$^{-1}$ | 10.0 |
| Zn | mg kg$^{-1}$ | 87.3 |
| Cu | mg kg$^{-1}$ | 56.6 |
| S | mg kg$^{-1}$ | 1950 |

*3.2. Dissolved P Removal by the Blind Inlet*

There was no trend between the raw flow rate and the raw inflow concentration of dissolved P ($p$ = 0.51, r$^2$ = −0.054, n = 145). In total, 65% and 82% of the cumulative dissolved P load were delivered to the blind inlet in the first and third events, respectively, after the single poultry litter application. Notice that after the third runoff event, which occurred within 50 days of poultry litter application, the inflow FWMC of dissolved P dramatically decreased (Table 4). Following application of manure or chemical fertilizer P, dissolved P concentrations in runoff are always elevated, followed by a decrease until the added P equilibrates with soil through adsorption and precipitation reactions [24,25]. Until that pseudo-equilibrium is achieved, dissolved P lost in runoff is dominated by the solubility of the P amendment itself and is referred to as "incidental" P loss [26]. Before the poultry litter application, inflow dissolved P concentrations were generally of the magnitude 0.1 to 0.2 mg L$^{-1}$ (data not shown), similar to events 4–7. Data in Table 4 illustrate the notion that incidental P loss events are less frequent but of greater magnitude per event compared to runoff events controlled by soil test P levels (i.e., "legacy P loss"). Data in Table 4 also highlight the fact that dissolved P loading in this case was influenced more by the dissolved P concentration rather than flow volume. For example, runoff event # 4 produced the largest volume, 1.6 times more water than event # 1, yet only delivered 6.6% of the cumulative dissolved P load. Except for incidental P losses such as events # 1–3 in this study, dissolved P loads are normally mostly controlled by flow volume [27].

**Table 4.** Dissolved phosphorus (P) removal, concentrations and loads at the inlet and outlet, and flow characteristics for each runoff event at the blind inlet.

| Date | Runoff Event # | Event Duration [a] | Peak Flow Rate | Total Discharge [b] | Dissolved P | | | | | |
|------|-----------|-----------------|---------------|------------------|-------------|------------|----------|----------|----------|----------|
| | | | | | FWMC [c] Inflow | FWMC Outflow | Mass in Inflow | Mass in Outflow | Removed [d] | |
| | | h | $m^3\,h^{-1}$ | $m^3$ | $mg\,L^{-1}$ | | g | | % | $mg\,kg^{-1}$ slag |
| 3 April | 1 | 12 | 26.4 | 256 | 2.719 | 1.307 | 695 | 334 | 52 | 23.9 |
| 15 April | 2 | 5 | 16.9 | 72 | 0.926 | 0.412 | 67 | 30 | 56 | 2.45 |
| 3 May | 3 | 12 | 27.6 | 187 | 0.611 | 0.140 | 114 | 26 | 77 | 5.83 |
| 22 June | 4 | 12 | 41.6 | 418 | 0.167 | 0.178 | 70 | 74 | −7 | −0.30 |
| 1 November | 5 | 12 | 26.7 | 159 | 0.335 | 0.236 | 53 | 35 | 33 | 1.18 |
| 26 November | 6 | 8 | 24.8 | 154 | 0.146 | 0.167 | 23 | 26 | −14 | −0.21 |
| 31 December | 7 | 12 | 24.4 | 241 | 0.173 | 0.239 | 42 | 58 | −38 | −1.06 |
| Total | | 73 | | 1487 | 0.715 | 0.394 | 1063 | 583 | 45 | 31.8 |

[a] Duration of runoff event, in hours. [b] Discharge of runoff event. [c] Flow-weighed mean concentration of dissolved P. [d] Dissolved P removed as percent, relative to the mass in the inflow; mg of dissolved P removed per kg of steel slag assuming 15,200 kg of steel slag in the blind inlet.

The cumulative mass of dissolved P delivered to the blind inlet over seven runoff events, from April to December 2018, was about 1 kg. In contrast, only 0.58 kg was detected in the outflow water, indicating that 45% of the dissolved P in the inflow water was removed when passed through the blind inlet. The blind inlet in this study was installed in late fall of 2015, but only events from 2018 are reported here due to prior sampling problems. Regardless of sampling, the blind inlet received P since its installation. Thus, the runoff events and dissolved P loading in addition to the 1 kg measured from the seven runoff events in 2018 resulted in an additional P removed beyond the 0.58 kg documented for 2018. As in any filtering system, the P removal efficiency decreases as a function of loading due to the consumption of reactants, which in this case is soluble Ca and the ability to buffer an elevated pH. For example, discrete P removal for the slag shown in Figure 2 illustrates how initial P removal is greatest and then decreases with further P additions.

Dissolved P removal by slag > 25 mm is negligible compared to the fine slag, due to the low Ca solubility and ability to buffer pH [21]. Therefore, P removal in this study was expressed per unit mass of the fine-sized EAF slag placed in the upper 0.3 m of the blind inlet (15.12 mg). The mass of PSM is important because it allows the calculation and normalization of P removal for comparison to other PSMs [8]. Thus, about 32 mg of dissolved P per kg of slag was removed from April to December 2018. The cumulative P removal by a relatively large size slag fraction in this study is comparable to both field and laboratory studies conducted on smaller EAF slag [19,21,22].

Dissolved P loads removed by the blind inlet are shown on a per-event basis in Table 4. About 75% of all dissolved P removed by the blind inlet occurred during the first event, and 100% occurred with the first three events. These results demonstrate that dissolved P removal was governed by events with the highest dissolved P concentrations in the inflow. When slag performance was expressed in terms of both cumulative P input and removal, Figure 3 shows that field data was better matched with laboratory flow-through experiments that utilized the 2.5 mg P $L^{-1}$ than the 0.2 mg P $L^{-1}$ inflow concentration. From Table 2, it is observed that events with inflow P concentrations less than 0.2 mg P $L^{-1}$ removed little to no dissolved P, but continued P loading at higher inflow concentrations resulted in further P removal. As previously discussed, this is due to the Ca phosphate precipitation mechanism by which slag removes dissolved P. Table 4 also reveals that three of the seven runoff events resulted in dissolved P release from the blind inlet. Under laboratory conditions, water extractions of spent slag from P removal structures released negligible P concentrations [9,28]. Thus, it is hypothesized that P-enriched sediments trapped by the blind inlet desorb/dissolve P when inflow concentrations decrease below the equilibrium P concentration for those P-saturated particles. For example, the three events that resulted in dissolved P release were also the events with the lowest inflow P concentrations (Table 4).

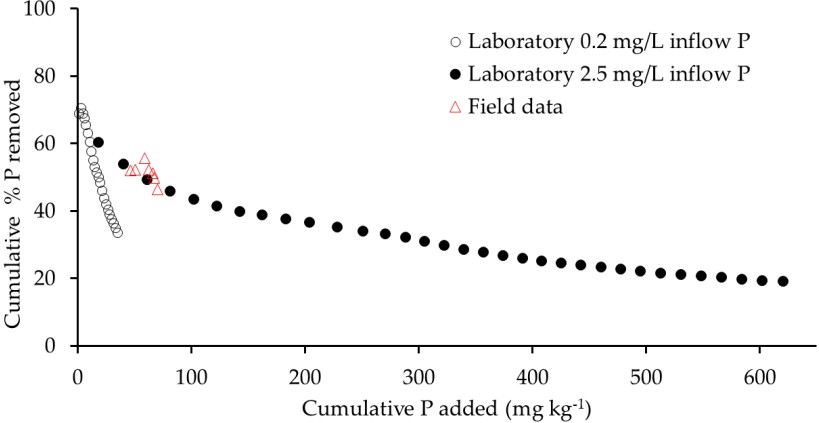

**Figure 3.** Cumulative dissolved phosphorus (P) removed by the slag under laboratory flow-through conditions (10 min retention time) at two different concentrations, compared to measured P removal in the blind inlet. Results expressed as a function of the cumulative P added per kg slag.

### 3.3. pH, Ca, C, N, K, Mg, and S

The pH of inflow water was lower than outflow water (Table 5), by an average of 0.95 pH units. The elevated pH of treated water is expected because of the chemical properties of the EAF slag used in this study (Table 3), which is in accord with the literature [19]. These chemical properties are important for removing dissolved P from water since Ca-phosphate precipitation is governed by the Ca concentration and pH [29,30]. The optimal pH for spontaneous Ca-phosphate precipitation is ≥ pH 8 [29,30]. In only two runoff events (# 6 and 7), the pH was <8, and no removal of dissolved P was observed. Overall, the pH of the inflow and outflow water decreased with time, from 7.33 to 6.84 in inflow water and from 8.44 to 7.17 in outflow water.

**Table 5.** Event and cumulative loads (in kg) in the inflow and outflow of the blind inlet for each runoff event.

| Analyte | Date | 3-April | 15-April | 3-May | 22-June | 1-November | 26-November | 31-December | Total |
| | Runoff Event # | 1 | 2 | 3 | 4 | 5 | 6 | 7 | |
|---|---|---|---|---|---|---|---|---|---|
| pH | Inflow | 7.33 | 8.19 | 6.64 | 7.32 | 6.69 | 6.92 | 6.84 | 7.05 [2] |
| | Outflow | 8.44 | 8.68 | 8.08 | 8.25 | 8.05 | 7.56 | 7.17 | 8.00 |
| | Pct Diff [1] | −15.1 | −6.0 | −21.7 | −12.7 | −20.2 | −9.2 | −4.9 | −13.5 |
| DOC [3] | Inflow | 6.13 | 1.57 | 2.65 | 3.51 | 2.79 | 2.36 | 3.81 | 22.8 |
| | Outflow | 6.66 | 1.47 | 3.91 | 5.97 | 3.16 | 3.23 | 5.15 | 29.6 |
| | Pct Diff | −8.7 | 6.0 | −47.4 | −69.9 | −13.4 | −36.8 | −35.5 | −29.6 |
| IC [3] | Inflow | 0.69 | 0.12 | 0.19 | 0.91 | 0.80 | 0.39 | 0.55 | 3.65 |
| | Outflow | 1.03 | 0.18 | 1.20 | 2.18 | 0.76 | 0.76 | 1.11 | 7.23 |
| | Pct Diff | −48.6 | −47.6 | −550 | −139 | 4.4 | −93.2 | −101 | −97.7 |
| NH$_4$-N | Inflow | 0.87 | 0.08 | 2.28 | 0.10 | 0.04 | 0.04 | 0.04 | 3.46 |
| | Outflow | 0.71 | 0.09 | 0.04 | 0.16 | 0.05 | 0.04 | 0.06 | 1.14 |
| | Pct Diff | 18.5 | −6.5 | 98.4 | −54.3 | −31.5 | −0.3 | −35.4 | 67.1 |
| NO$_3$-N | Inflow | 0.61 | 0.60 | 0.86 | 0.57 | 0.40 | 0.19 | 0.39 | 3.62 |
| | Outflow | 0.65 | 0.56 | 1.26 | 0.80 | 0.66 | 0.21 | 0.39 | 4.52 |
| | Pct Diff | −7.5 | 7.1 | −45.2 | −38.6 | −67.1 | −10.4 | −1.9 | −25.1 |
| Org N [3] | Inflow | 2.17 | 0.13 | 0.39 | 0.41 | 0.18 | 0.25 | 0.35 | 3.89 |
| | Outflow | 1.69 | 0.13 | 0.12 | 0.40 | 0.21 | 0.22 | 0.43 | 3.20 |
| | Pct Diff | 22.2 | 3.6 | 69.8 | 2.5 | −13.6 | 12.9 | −22.5 | 17.9 |
| TP [3] | Inflow | 1.38 | 0.09 | 0.27 | 0.34 | 0.14 | 0.15 | 0.19 | 2.54 |
| | Outflow | 0.93 | 0.06 | 0.15 | 0.33 | 0.11 | 0.13 | 0.28 | 1.98 |
| | Pct Diff | 32.3 | 31.2 | 45.4 | 1.9 | 21.1 | 13.4 | −47.7 | 22.0 |
| K | Inflow | 2.33 | 0.49 | 10.30 | 2.55 | 2.33 | 2.14 | 4.95 | 25.1 |
| | Outflow | 2.01 | 0.48 | 1.85 | 2.93 | 2.19 | 1.89 | 3.94 | 15.3 |
| | Pct Diff | 13.5 | 2.7 | 82.0 | −14.8 | 5.9 | 11.9 | 20.4 | 39.1 |
| Mg | Inflow | 1.15 | 0.44 | 1.76 | 0.82 | 0.88 | 0.59 | 0.84 | 6.47 |
| | Outflow | 0.60 | 0.23 | 0.74 | 0.29 | 0.34 | 0.24 | 0.40 | 2.85 |
| | Pct Diff | 47.8 | 48.9 | 57.7 | 64.3 | 60.6 | 58.6 | 52.5 | 56.0 |
| Ca | Inflow | 6.61 | 2.57 | 10.94 | 5.12 | 3.93 | 2.67 | 3.37 | 35.2 |
| | Outflow | 11.61 | 5.51 | 15.56 | 10.43 | 7.45 | 6.20 | 7.60 | 64.4 |
| | Pct Diff | −75.5 | −115 | −42.2 | −104 | −89.5 | −132 | −126 | −82.8 |
| SO$_4$-S | Inflow | 0.79 | 0.21 | 4.64 | 1.50 | 0.30 | 0.19 | 0.25 | 7.88 |
| | Outflow | 0.67 | 0.21 | 2.39 | 1.71 | 0.32 | 0.21 | 0.49 | 6.00 |
| | Pct Diff | 14.9 | 2.1 | 48.5 | −13.8 | −6.6 | −10.9 | −99.8 | 23.8 |

[1] Pct Diff: Percent difference; a negative number indicates an increase, whereas a positive number indicates a decrease in the outflow relative to the inflow. [2] Average pH. [3] DOC: Dissolved organic C; IC: total inorganic C; TKN: NH4-N + organic N; Org N: organic N; TP: Total P.

The cumulative mass of Ca from the inflow and outlet water was 35.2 and 64.4 kg, respectively. In all runoff events, the mass of Ca was higher in the outflow than in the inflow water (Table 5). The largest outflow Ca mass was observed for runoff events # 1, 3, and 4 (10.4 to 15.6 kg). However, average raw

Ca concentrations were lowest for runoff events # 4, 6, and 7 (from 26.7 to 39.5 mg L$^{-1}$), which coincide with the events that did not result in dissolved P removal. This is expected since it has been suggested that for Ca-phosphate precipitation to occur at pH 8, Ca concentrations between 30 and 100 mg L$^{-1}$ are necessary; at least 50 mg Ca L$^{-1}$ is required for complete dissolved P removal [29]. For a tile-drain P removal structure utilizing EAF slag, dissolved P removal dramatically decreased when the pH of treated water dropped to 8.5 or less [31]. In addition, Ca phosphate precipitation is also subject to kinetics, and therefore favored by a greater retention time [7]. These results confirm that a combination of pH and the Ca and P concentrations partly control the removal of dissolved P from the inflow water. For example, considering outflow water from runoff event # 2, pH was 8.68 with an average Ca concentration of 77.1 mg L$^{-1}$ and dissolved P removal of 77.2%, compared to runoff event # 4, where pH was 8.25 with an average Ca concentration of 26.7 mg L$^{-1}$, and no removal of dissolved P occurred.

Other analytes determined in water samples included K, Mg, Ca, and sulfate-S. The cumulative K (25.1 kg), Mg (6.47 kg), and sulfate-S (7.88 kg) contents in the inflow water were reduced by 39.1%, 56.0%, and 23.8%, respectively, in the outflow water. In total, 27% to 59% of the total inflow for each analyte above was accounted for in the 3 May runoff event. Similar trends were observed for the outflow water, except for K, where 26% of the total was accounted for in the last runoff event (31 December). Because of alkalinity, Mg concentrations may have been decreased due to precipitation with carbonates, which readily form from DIC under alkaline conditions. Sulfate may have been precipitated as gypsum (Ca sulfate) due to the slag serving as a Ca source; gypsum formation in EAF slag columns has been observed [18].

### 3.4. Carbon, Total Phosphorus, and Nitrogen

The TDC content was 26.5 and 36.8 kg for the inflow and outflow water, respectively, of which, 86.2% and 80.4% of TDC were accounted for by DOC in the inflow and outflow water, respectively. In this study, a DOC net loss of 6.74 kg was observed, suggesting that the blind inlet bed was a source of DOC. The blind inlet was receiving runoff for two years before the data collection reported in this study; thus, DOC accumulated before 2018. Fresh EAF slag is rich in carbonates (i.e., DIC) but devoid of DOC due to the lack of organic C inputs and excessive temperature of the steelmaking process. Sources of organic C in the inflow water include the runoff C from applied poultry litter and sediments. The highest inflow DOC (6.13 kg) was observed in the first runoff event, which occurred 19 days after poultry litter application, whereas in the other runoff events, DOC content ranged from 1.57 to 3.91 kg. These results are not surprising, given that manure runoff losses are affected by the timing of manure application relative to runoff events [32,33]. Outflow DOC was higher than inflow, except in runoff event # 2, which had the lowest DOC content for both inflow and outflow. The higher DOC in the outflow relative to inflow may be explained in part by the dissolution of retained organic C in the blind inlet when in contact with alkaline EAF slag during runoff events. Contact between runoff water and EAF slag of high pH (Table 3) may dissolve the deposited organic C in the blind inlet. It is common knowledge that organic C dissolves at high pH [34,35].

Interestingly, in addition to the first runoff event influenced by poultry litter application, other events had high DOC, e.g., # 4 on 22 June (5.97 kg) and # 7 on 31 December (5.15 kg). Runoff sediments may explain the high DOC during runoff event # 4 due to soil disturbance during planting, which occurred 28 days before runoff event # 4 (Table 1). The high DOC during runoff event # 7 could be due to the residue runoff and decomposition after harvest, which occurred 26 days before the runoff event (Table 1).

Inherently, EAF slag contains inorganic C (IC); thus, it not surprising to observe higher IC content in outflow than inflow water (Table 5). In runoff event # 4 (22 June), the outflow IC was at least double the amount of other runoff events (2.18 kg vs. 0.18 to 1.20 kg). As observed for DOC, planting occurred 28 days before runoff event # 4; thus, we hypothesize that DOC from runoff sediments interacted with soluble Ca in EAF slag, releasing IC, and potentially reduced Ca-phosphate

precipitation. Calcium–organic C interactions through cation bridging are known mechanisms to stabilize DOC in the environment [36–38].

Total P (TP) in all runoff events was higher in the inflow water (2.54 kg) than in the outflow water (1.98 kg), suggesting that the slag material removed 22% of the TP from the inflow water (Table 5). The highest TP content was observed in the first runoff event for both inflow and outflow (54% and 47% of the cumulative TP load, respectively). Total P is the sum of dissolved P and particulate P. Filtration of sediment is known to reduce particulate P, which was one of the original purposes of the blind inlet [4]. Over a 12-year lifespan, a blind inlet constructed with limestone gravel was able to remove 40% of sediment and particulate P [4].

Cumulative organic N content was higher in the inflow water (3.89 kg) than outflow (3.20 kg), suggesting that the blind inlet was a sink for organic N, reducing organic N losses by 18% (Table 5). The highest organic N content for both inflow and outflow was observed in the first runoff event, accounting for 56% and 53% of the cumulative organic N, respectively, again indicating the impact of the poultry litter application. Only in two runoff events (1 November and 31 December) was organic N higher in the outflow than inflow, suggesting that the blind inlet served as a source of organic N; however, the organic N from the inflow water of these two events only accounted for 14% of cumulative organic N.

The cumulative nitrate-N content in the outflow (4.52 kg) was 25% higher than inflow water, suggesting that the blind inlet in this study was a source of nitrate-N. Conversely, cumulative ammonium-N (3.46 kg) from the inflow was 67% higher than the outflow, suggesting that the blind inlet served as a sink for ammonium-N (Table 5). About 25% and 66% of the cumulative ammonium-N from inflow water was accounted for by runoff events on 3 April and 3 May, respectively. In all, the cumulative N fraction (nitrate + ammonium + organic N) in the inflow was 11 kg compared to the outflow water with 8.86 kg, suggesting that the blind inlet reduced the losses of total N by 19.3%. The release of nitrate and ammonium from the blind inlet is likely due to N cycling occurring within the blind inlet, similar to the release of DOC. Since the blind inlet retains sediments, as indicated by TP removal as a proxy, organic-rich sediments are expected to mineralize and nitrify when conditions are suitable.

*3.5. Herbicides*

From April to December 2018, three herbicides (glyphosate, dicamba, and atrazine) were applied to the site (Table 1). Therefore, these chemicals and their metabolites (AMPA from glyphosate and OH-ATZ, DEA, and DIP from atrazine) were quantified in inflow and outflow samples (Table 6). The cumulative contents of glyphosate and its metabolite AMPA in the inflow were relatively low (0.062 and 0.249 g, respectively) compared to the amount of glyphosate typically applied in a single application, which only occurred 2 and 15 days before runoff events on 3 May and 22 June, respectively. These results suggest that glyphosate is strongly held in soils, and small amounts are found in runoff, which is in accordance with the literature [39,40]. The higher cumulative content of AMPA in the inflow compared to glyphosate might be due to the higher water solubility of the former than the parent compound and the relatively short half-life (24 days) of glyphosate under field conditions [41]. Furthermore, the cumulative contents of glyphosate and AMPA in the outflow were 73.6% and 96.2% lower, respectively, relative to inflow, suggesting that the steel slag or sediment retained within the blind inlet reduced losses of these compounds. Glyphosate and AMPA have a phosphonate moiety, which is thought to have similar sorption mechanisms as phosphate in soils [42–44], i.e., coordination bonding with cations, including Ca, Fe, and Al [42]; thus, we hypothesize that glyphosate and AMPA are removed from water by the EAF slag in the blind inlet through coordination bonding with cations present in the EAF slag. Due to the relatively short retention time of the blind inlet (~7 min), it is unlikely that removal was due to biological reactions.

**Table 6.** Event and cumulative loads of several herbicides and their respective metabolites (in mg) in the inflow and outflow of the blind inlet.

| Analyte | Date | 3-April | 15-April | 3-May | 22-June | 1-November | 26-November | 31-December | Total |
|---|---|---|---|---|---|---|---|---|---|
| | Runoff Event # | 1 | 2 | 3 | 4 | 5 | 6 | 7 | |
| ATZ [1] | Inflow | 0 | 0 | 48 | 3661 | 1 | 3 | 2 | 3716 |
| | Outflow | 1 | 0 | 5 | 5828 | 6 | 18 | 2 | 5860 |
| | Pct Diff [2] | −153 | 97 | 89 | −59 | −371 | −560 | 1 | −58 |
| OH-ATZ [1] | Inflow | 22 | 4 | 6 | 1259 | 40 | 31 | 35 | 1398 |
| | Outflow | 21 | 5 | 36 | 2631 | 34 | 29 | 47 | 2803 |
| | Pct Diff | 5 | −26 | −496 | −109 | 16 | 7 | −33 | −100 |
| DIP [1] | Inflow | 2 | 1 | 37 | 2684 | 1 | 0 | 1 | 2724 |
| | Outflow | 1 | 1 | 6 | 2091 | 0 | 2 | 0 | 2101 |
| | Pct Diff | 32 | 7 | 83 | 22 | 100 | N/A [3] | 65 | 23 |
| DEA [1] | Inflow | 2 | 1 | 42 | 2435 | 5 | 5 | 5 | 2494 |
| | Outflow | 1 | 1 | 4 | 2240 | 5 | 4 | 5 | 2260 |
| | Pct Diff | 45 | −14 | 90 | 8 | 3 | 10 | 5 | 9 |
| ATZ-MET [1] | Inflow | 26 | 5 | 85 | 6378 | 46 | 36 | 41 | 6617 |
| | Outflow | 23 | 6 | 47 | 6962 | 39 | 35 | 52 | 7164 |
| | Pct Diff | 10 | −21 | 45 | −9 | 15 | 3 | −27 | −8 |
| TRIA [1] | Inflow | 26 | 6 | 133 | 10,039 | 47 | 39 | 43 | 10,333 |
| | Outflow | 24 | 6 | 52 | 12,790 | 45 | 53 | 54 | 13,024 |
| | Pct Diff | 7 | −11 | 61 | −27 | 5 | −37 | −26 | −26 |
| GLY [1] | Inflow | 0 | 0 | 52 | 10 | 0 | 0 | 0 | 62 |
| | Outflow | 2 | 0 | 0 | 14 | 0 | 0 | 0 | 16 |
| | Pct Diff | −1022 | −305 | 100 | −39 | N/A | N/A | N/A | 74 |
| AMPA [1] | Inflow | 3 | 1 | 224 | 2 | 0 | 19 | 0 | 249 |
| | Outflow | 1 | 0 | 1 | 7 | 1 | 0 | 1 | 9 |
| | Pct Diff | 81 | 100 | 100 | −242 | −210 | 100 | # N/A | 96 |
| GLY+ [1] | Inflow | 3 | 1 | 275 | 12 | 0 | 19 | 0 | 311 |
| | Outflow | 3 | 0 | 1 | 20 | 1 | 0 | 1 | 26 |
| | Pct Diff | 18 | 64 | 100 | −72 | −254 | 100 | # N/A | 92 |
| Dicamba | Inflow | 4 | 0 | 76 | 0 | 0 | 0 | 0 | 79 |
| | Outflow | 0 | 0 | 0 | 0 | 0 | 0 | 5 | 5 |
| | Pct Diff | 100 | N/A | 100 | N/A | N/A | N/A | N/A | 94 |

[1] ATZ: Atrazine; OH-ATZ: 2-hydroxyatrazine' DIP: desisopropyl atrazine; DEA: desethyl atrazine; ATZ-MET: sum of atrazine metabolites (OH-ATZ, DIP, and DEA); TRIA: Sum of atrazine + metabolite; GLY: glyphosate; AMPA: aminomethylphosphonic acid, a glyphosate metabolite; GLY+: glyphosate + AMPA. [2] Pct Diff: Percent difference; a negative number indicates an increase, whereas a positive number indicates a decrease in the outflow relative to the inflow. [3] N/A: Not available.

Dicamba was applied in a single application, 2 days before the runoff event on 3 May; however, like glyphosate, the cumulative load was low (0.079 and 0.005 g for the inflow and outflow, respectively). Dicamba is a benzoic acid herbicide with pKa of 1.87, low soil sorption coefficients, and a short half-life in the field (~4 days) [41]; thus, this herbicide is not expected in runoff at large quantities. If dicamba is present in runoff, it will be present in its deprotonated form; thus, it could be removed by cation coordination like glyphosate and AMPA. Regardless of the mechanism, about 95% of the cumulative dicamba from the inflow was not accounted for in outflow water.

The cumulative atrazine content in the outflow water (5.86 g) was 57.7% higher than inflow water, suggesting that the blind inlet in this study behaved as a temporary storage for atrazine previously deposited via sediments. Sorption of atrazine is governed by several factors, including organic C, clay mineralogy, and pH [11]. Furthermore, 98.5% and 99.5% of the cumulative atrazine load measured in the inflow and outflow, respectively, were accounted for in the 22 June runoff event. The cumulative load of the three atrazine metabolites was 1.8 and 1.2 times higher than atrazine measured in the inflow and outflow water, respectively. The content of these metabolites in the 22 June runoff event, like atrazine, accounted for 96.4% and 97.2% of the cumulative inflow and outflow load, respectively. The

cumulative amounts of DEA and DIA metabolites were 22.9% and 9.4% higher in the inflow, relative to the outflow water, respectively. However, the cumulative amount of OH-ATZ was 100.5% higher in the outflow water, relative to the inflow water. Altogether, the cumulative atrazine + metabolites were 10.33 and 13.02 g from the inflow and outflow water, respectively, of which >97% was accounted for in the 22 June runoff event. The highest content of atrazine and its metabolites from this single event is explained by the fact that atrazine was applied 15 days before the runoff event occurred. Surface transport of atrazine is higher, with shorter time intervals between application and runoff events [45,46]. Atrazine metabolites produced by N-dealkylation reactions (i.e., DEA and DIA) are considered products of biodegradation [47], and the half-life of atrazine under field conditions is about 26 days [41]. In contrast, the dechlorination reaction of atrazine (i.e., OH-ATZ metabolite) can be either abiotic or biotic, with the former (via hydrolysis) being widely accepted [47,48]. Hydrolysis of atrazine is governed by pH; under acidic and alkaline conditions, hydrolysis of atrazine increases [48]. Atrazine binds to soil particles (i.e., particulate soil-bound atrazine); however, in this study, only the dissolved form of atrazine was quantified. In blind inlets, herbicides like atrazine and its metabolites might be retained as sediment bound, or dissolved atrazine could sorb to previously deposited sediments [2], although sediment-bound atrazine was unaccounted for in this study. Slag does not sorb atrazine [6], but it may promote the desorption of sediment-bound atrazine trapped in the blind inlet due to changes in the chemical environment, including increases in pH and solubilization of sediment-bound organic C due to the alkaline nature of slag. Such a mechanism provides a hypothesis for greater dissolved atrazine and its metabolites in the outflow than inflow. Conversely, with a limestone-bed blind inlet, atrazine losses were reduced by 57%, relative to the tile riser, which was attributed to the accumulation of organic C and near-neutral pH in the blind inlet (~7.6) [2]. The high pH of EAF slag may catalyze the hydrolysis of atrazine to its dechlorinated metabolite, OH-ATZ. This confirms the importance of pH conditions and accumulation of organic C for atrazine removal.

## 4. Summary and Implications

Although the slag-filled blind inlet in this study had been receiving P for approximately two years before reliable inflow sample collection, it still removed 45% of the cumulative inflow dissolved P from April to December 2018 (about 0.5 kg). Although it is unknown how much dissolved P flowed into the blind inlet before April 2018, it can be assumed that the P removal efficiency at that time was greater than what was measured during 2018, simply due to the nature of chemical filtration and P removal by slag. Both laboratory flow-through tests and field observations confirmed the notion that EAF slag is more efficient at removing dissolved P at higher inflow concentrations, e.g., 2.5 vs. 0.2 mg $L^{-1}$, concomitant with the Ca phosphate precipitation mechanism. Most of the dissolved P removed during the monitoring period occurred in only three of the seven events, which had the highest inflow P concentrations that resulted from a recent poultry litter application to the study field. Dissolved P removal efficiency was also influenced by the pH and Ca concentration of the treated water, where increases in both promoted removal via Ca phosphate precipitation.

If dissolved P loading to the blind inlet for the previous two years is extrapolated from the third year that was sampled, then the cumulative 3-year loading would be around 200 mg $kg^{-1}$. Then, if the majority of this loading occurred via higher dissolved P concentrations as observed in this study (i.e., ~2.5 mg $L^{-1}$), the cumulative dissolved P removed over the 3-year period would be approximately 36%, or 1 kg, based on the laboratory flow-through P removal curve with a 2.5 mg $L^{-1}$ inflow P concentration. At low inflow dissolved P concentrations of 0.2 mg $L^{-1}$ or less, little to no further dissolved P removal is expected. However, at an inflow concentration of 2.5 mg $L^{-1}$, the slag would continue to remove dissolved P for over 10 years, assuming an annual inflow load of 36 mg $kg^{-1}$. Currently, the blind inlet continues to be effective after three years.

The blind inlet removed 22% of the cumulative TP load (which includes particulate P). This value is likely an underestimate since the inflow sampling was not designed to capture suspended sediment effectively, and the focus of this study was to determine the impact of the slag-blind inlet on

dissolved P removal. The blind inlet also removed total N, generally in the form of organic N and ammonium. However, it appeared that N cycling was occurring within the slag bed, since organic N and ammonium decreased, yet nitrate loads increased from the blind inlet. Since organic N is mostly transported in particulate form, trapped sediment will undergo N transformations, i.e., mineralization and nitrification in between flow events. Dicamba, glyphosate, and its metabolite, AMPA, were removed by the blind inlet by over 70% based on loads.

While the blind inlet removed several herbicides and nutrients, the measured outflow loads of atrazine, its metabolites, and DOC were greater than inflow loads during the monitoring period. It is important to keep in mind that only dissolved forms of these compounds were measured, not particulate forms. Therefore, since there is no inherent DOC or atrazine found in new EAF slag material, particulate forms of DOC and particulate-bound atrazine associated with sediments appeared to have been delivered into the blind inlet, where abiotic and biotic reactions solubilized such forms, allowing them to be detected in greater concentrations in the outflow water.

The timing of nutrient and herbicide delivery was strongly controlled by field management. Specifically, the majority of P, N, and DOC delivered to the blind inlet occurred in a single runoff event, which was only 19 days after the application of poultry litter to the source field. Similarly, almost all herbicides delivered to the blind inlet occurred in a single runoff event that coincided with the application of those herbicides to the field. These observations highlight the importance of runoff timing relative to field management. Producers can appreciably minimize the risk of contaminant transport by avoiding application at times when large rainfall events capable of producing runoff are expected to occur.

At a cost of 1500 USD, the installation of the blind inlet using steel slag did not cost more than conventional blind inlets constructed with local aggregate, such as limestone. While limestone-based blind inlets have been shown to remove little to no dissolved P [4,5], this study demonstrated that a simple update to slag would permit blind inlets to remove dissolved P in addition to sediment, particulate P, and provide obstruction-free drainage of depressions. The slag used in this study did not clog up, continues to remove dissolved P after three years, and performed as expected based on laboratory flow-through experiments, unlike the slag P removal structure reported by Penn et al. [31]. The main difference in those studies is the source water; this study treated surface runoff water instead of tile drainage water, which contains appreciable bicarbonate and dissolved forms of $CO_2$ compared to surface runoff [31]. The under-performance of slag in their study was attributed to the formation of Ca carbonate instead of Ca phosphate through bicarbonate input, which additionally clogged the structure, besides consuming soluble Ca that would usually precipitate Ca phosphate for P removal. Based on the results of this and other studies that utilized EAF slag to treat surface runoff [10,19], EAF slag appears to perform as predicted under laboratory flow-through testing, unlike P removal structures constructed to treat tile drainage water.

**Author Contributions:** Conceptualization, J.M.G., C.J.P., S.J.L.; methodology, J.M.G., C.J.P., S.J.L.; writing—original draft preparation, J.M.G.; writing—review and editing, C.J.P., S.J.L. All authors have read and agreed to the published version of the manuscript.

**Funding:** This research received no external funding.

**Acknowledgments:** We would like to thank Levy Corp. for providing the steel slag and Bowman & Bowman Farms for allowing installation of the blind inlet on their property.

**Conflicts of Interest:** The authors declare no conflict of interest.

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
