# Peer review of "Utilization of Steel Slag in Blind Inlets for Dissolved Phosphorus Removal"

_water, doi:10.3390/w12061593_

Round 1
Reviewer 1 Report
This is very well written manuscript. In addition, the topic is very imortant as pollution delivered through tile draiange may seriously deteriorate surface waters. I did not find any flaws or shortcomings.
My only concern is Figure 1. It is very difficult to understand what is the water flow. Authors should provide a simple drawing showing water movement through the system.
Author Response
Thank you for the comments. Figure 1 was modified to show the flow of water (blue arrows).
Reviewer 2 Report
Water, Peer Review 819405
“Utilization of steel slag in blind inlets for dissolved phosphorus removal”
General comments:
The paper submitted for review is an interesting case study of using steel slag to control nutrients in drainage water. Despite the fact that the overall sorption capacity of the tested material in relation to phosphorus is moderately low (several dozen mg / kg), the idea of managing steel slag as fillings for blind inlets is worth supporting. The authors showed that it is an interesting alternative to commonly used limestone. The work is written clearly, concisely and exhaustively describing the research methodology and results. I believe that the work does not require substantive corrections, but I have some minor comments.
Abstract
The abstract is clearly written, stating the purpose, general methods, and results of the study.
Introduction
In this chapter, the authors could supplement information on how to handle filter material deposits after their life cycle.
Material and Methods
This chapter is the strength of the work. Research methods are described very accurately and logically.
l. 115. It seems to me that millimeter accuracy is exaggerated. The material is a porous aggregate, so it is enough to express this accuracy using the phrase "about 0.05m".
l. 141. In the subsection "Chemical analysis of water samples" you should solve the problem of the lost symbol "micro".
Results and discussion
l. 251. In table 3, the authors present the results of analyzes of the material with specific granulation (fine). In earlier chapters (e.g. line 92-94) they describe their experiments as research conducted on two types of material (fine and coarse). So there is an inaccuracy and it is worth explaining.
The article presented for review is a valuable contribution to the development of methods to protect waterbodies from nutrient runoff along with drainage water. Research has application potential because the problem of controlling non-point sources of pollution has become very important in recent decades. Phosphorus is recognized as a major factor in the degradation of natural aquatic ecosystems. It is increasingly important to develop effective methods for neutralizing the runoff of this element from non-point sources. Thus, the research objective responds well to environmental needs and policies. The article is written in a clear scientific language, contains accurate descriptions of experiments and clearly reported results. The conclusions are correct. The studies undertaken are not groundbreaking, but their results can contribute to the improvement of water protection practices in catchments. The work can be published in the Water journal.
Reviewer 3 Report
Very relevant and interesting work.
But it could be improved.
Research Manuscript Sections wrong. There should be Results and Discussion separately (https://www.mdpi.com/journal/water/instructions), Conclusions but not Summary and Implications.
The titles of the sections 3.3 to 3.5 of the results should be corrected, otherwise the table Mendeleev should be quoted. Could be "Evaluation of pH, Ca, C, N, K, Mg, and S concentrations"
Lack of mathematical statistical processing, too much "green" data.
